# Extract of *Acanthopanax senticosus* and Its Components Interacting with Sulfide, Cysteine and Glutathione Increase Their Antioxidant Potencies and Inhibit Polysulfide-Induced Cleavage of Plasmid DNA

**DOI:** 10.3390/molecules27175735

**Published:** 2022-09-05

**Authors:** Anton Misak, Marian Grman, Lenka Tomasova, Ondrej Makara, Miroslav Chovanec, Karol Ondrias

**Affiliations:** 1Institute of Clinical and Translational Research, Biomedical Research Center, Slovak Academy of Sciences, Dubravska Cesta 9, 845 05 Bratislava, Slovakia; 2Forest Arboretum Liptovsky Hradok, Hradna 534, 033 01 Liptovsky Hradok, Slovakia; 3Department of Genetics, Cancer Research Institute, Biomedical Research Center, Slovak Academy of Sciences, Dubravska Cesta 9, 845 05 Bratislava, Slovakia

**Keywords:** *Acanthopanax senticosus*, ^●^cPTIO, radical, antioxidant, H_2_S, polysulfides, DNA cleavage, caffeic acid, chlorogenic acid, eleutheroside

## Abstract

Aqueous root extract from *Acanthopanax senticosus* (ASRE) has a wide range of medicinal effects. The present work was aimed at studying the influence of sulfide, cysteine and glutathione on the antioxidant properties of ASRE and some of its selected phytochemical components. Reduction of the 2-(4-carboxyphenyl)-4,5-dihydro-4,4,5,5-tetramethyl-1H-imidazol-1-yloxy-3-oxide (^●^cPTIO) stable radical and plasmid DNA (pDNA) cleavage in vitro assays were used to evaluate antioxidant and DNA-damaging properties of ASRE and its individual components. We found that the interaction of ASRE and its two components, caffeic acid and chlorogenic acid (but not protocatechuic acid and eleutheroside B or E), with H_2_S/HS^−^, cysteine or glutathione significantly increased the reduction of the ^●^cPTIO radical. In contrast, the potency of ASRE and its selected components was not affected by Na_2_S_4_, oxidized glutathione, cystine or methionine, indicating that the thiol group is a prerequisite for the promotion of the antioxidant effects. ASRE interacting with H_2_S/HS^−^ or cysteine displayed a bell-shaped effect in the pDNA cleavage assay. However, ASRE and its components inhibited pDNA cleavage induced by polysulfides. In conclusion, we suggest that cysteine, glutathione and H_2_S/HS^−^ increase antioxidant properties of ASRE and that changes of their concentrations and the thiol/disulfide ratio can influence the resulting biological effects of ASRE.

## 1. Introduction

*Acanthopanax senticosus* (Rupr. et Maxim.) Maxim. (AS), synonymous with *Eleutherococcus senticosus,* and its extract are widely used in countries such as China, Korea, Japan and Russia for specific pharmacologic effects. It contains various phytochemical compounds that ensure its broad-spectrum effects. The numerous and diverse pharmacological activity of AS and its individual components involve for example, anti-tumour, anti-inflammatory, anti-radiation and cardiovascular protection potencies. In addition, AS positively influences the central nervous system, prevents and treats respiratory infections and improves physical fatigue effects [1,2,3]. AS extracts, as adaptogens, increase adaptability and survival of organisms under stress, where observed antioxidant properties can positively contribute to the effects [2,4,5,6,7,8]. Polysaccharides, flavones and several dozen individual compounds were isolated from AS having numerous biological effects [2,8]. The AS component chlorogenic acid (CGA) is a naturally occurring nonflavonoid polyphenol that exerts numerous biological effects, e.g., antiviral, antitumor, antibacterial, antioxidant, antiinflammatory, antilipidemic, antidiabetic, antihypertensive and antimicrobial activity [9,10,11,12,13]. Similarly, the AS component caffeic acid (CA) is a naturally occurring polyphenol in the human diet with antioxidant, anti-inflammatory, antimicrobial, cytostatic, hepatoprotective, neuroprotective, antidiabetic and positive cardiovascular properties [14,15,16,17]. Molecular mechanisms of CGA and CA effects involve, e.g., the reduction of reactive oxygen species, modulation of activities of several enzymes and numerous molecular pathways [5,9,10,11,12,13,14,15,16,17,18,19,20]. However, the exact molecular mechanisms in many of CGA or CA effects are still not fully understood.

Endogenously produced H_2_S/HS^−^ and polysulfides (H_2_S_n_) affect many physiological and pathological processes, such as hypertension, atherosclerosis, heart failure, inflammation, neurodegenerative diseases, cancer, etc. H_2_S/HS^−^ and H_2_S_n_ have the beneficial effects under conditions of oxidative stress by reacting with reactive oxygen species (ROS) [21,22,23,24,25]. They can interact with other cellular components, and products of these interactions have additional biological effects [22,26,27,28]. It has been observed that the products of the interaction of H_2_S/HS^−^ with NO, selenite, phthalic selenoanhydride and doxycycline have pronounced antioxidant effects [28,29,30,31].

Both sulfur species and the components of AS extract influence several “stress” biological processes, but the protective mechanisms are still not fully understood [7,21,22,23,24,25]. One of the key activities of these compounds is the scavenging of radicals. Therefore, we aimed to study a consequence of their mutual interaction in terms of affecting the stability of the 2-(4-carboxyphenyl)-4,5-dihydro-4,4,5,5-tetramethyl-1H-imidazol-1-yloxy-3-oxide (^●^cPTIO) stable radical and integrity of the plasmid DNA (pDNA). The present work was aimed at better understanding of the antioxidant effects of AS and some of its individual components either alone or mixed with biologically active sulphur species.

## 2. Results

### 2.1. Experiments Using ASRE

#### 2.1.1. ASRE Interacting with H_2_S Reduces ^●^cPTIO

Na_2_S used in our study dissociates in aqueous solution and reacts with H^+^ to yield H_2_S, HS^−^, and a trace of S^2−^. We use the term H_2_S to describe the total mixture of H_2_S, HS^−^, and S^2−^ forms. Since H_2_S is endogenously produced in organisms and exogenous H_2_S donors are being considered to be used in medicine, we have studied the interaction of H_2_S with ASRE and the ability of the generated products to reduce the ^●^cPTIO stable radical. The UV-VIS spectrum of ASRE has a relevant absorbance (ABS) peak at <400 nm (Figure 1A). In our study, the concentration of 75 µg mL^−1^ ASRE had an absorbance (ABS) at 322 nm equal to 0.6. The ASRE/^●^cPTIO mixture had an additional ABS peak of ^●^cPTIO at 560 nm (Figure 1A). During reduction of ^●^cPTIO, peaks at 358 and 560 nm decreased (Figure 1B). Since the peak at 560 nm did not significantly interfere with the spectrum of ASRE or its individual components, it was used to study the antioxidant properties of ASRE and its selected individual components interacting with H_2_S, Na_2_S_4_, Cys, GSH, glutathione oxidized (GSSG), cystine and methionine (MET).

Radical ^●^cPTIO (100 µmol L^−1^) was stable in the phosphate buffer and H_2_S (10–400 µmol L^−1^) did not decrease the ^●^cPTIO concentration for at least 30 min (Figure 2A,B) as demonstrated by a negligible ABS decrease at 560 nm. ASRE had a moderate effect in terms of decreasing the concentration of ^●^cPTIO (decrease in concentration of ^●^cPTIO reflects its reduction) (Figure 2A). However, addition of H_2_S (≥5 µmol L^−1^) to the ASRE/^●^cPTIO mixture significantly increased the reduction of ^●^cPTIO in a concentration-dependent manner (Figure 2A). Similarly, an increase of ASRE concentration (≥6 µg mL^−1^) in the ^●^cPTIO/ASRE/H_2_S mixture significantly increased the reduction of ^●^cPTIO (Figure 2B). These results indicate that the interaction of H_2_S with ASRE led to the reduction of ^●^cPTIO. The potency of ASRE and the ASRE/H_2_S mixture to reduce ^●^cPTIO was also compared with that of (±)-6-hydroxy-2,5,7,8-tetramethylchromane-2-carboxylic acid (Trolox), a water-soluble derivative of α-tocopherol (Figure 2C): for example, the potency of 150–300 µg mL^−1^ ASRE (Figure 3B,C) was comparable with the potency of 100–200 µmol L^−1^ Trolox (Figure 2C), or the potency of the mixture of 150 µg mL^−1^ ASRE with 10 µmol L^−1^ H_2_S (Figure 2A) was comparable with that of 200–400 µmol L^−1^ Trolox (Figure 2C).

#### 2.1.2. ASRE Interacting with Cys or GSH (But Not with GSSG, MET or Cystine) Reduces ^●^cPTIO

Since H_2_S increased the reduction of ^●^cPTIO in the presence of ASRE, Cys and GSH (having a thiol group) were subsequently studied as well. Cys (400 µmol L^−1^) alone had a minor effect on ^●^cPTIO reduction (Figure 3A). Similarly to H_2_S, the addition of ASRE to Cys significantly increased the reduction of ^●^cPTIO, with the resulting effect being strongly ASRE-dose-dependent at a constant Cys/^●^cPTIO ratio (Figure 3A). However, at a constant ASRE/^●^cPTIO ratio, saturation at a higher Cys concentration was reached (Figure 3B). Next, the ability of GSH, GSSG, MET and cystine to reduce ^●^cPTIO was evaluated. GSH (400 µmol L^−1^) on its own did not reduce ^●^cPTIO (Figure 3C). However, the addition of ASRE to the GSH/^●^cPTIO mixture caused the reduction of this radical. In contrast, the mixture of GSSG, MET or cystine (400 µmol L^−1^) with ASRE had only a negligible effect on ^●^cPTIO reduction (Figure 3C). Potency of the compounds to increase the ^●^cPTIO-reducing effect of ASRE was in the order of: H_2_S/ASRE > Cys/ASRE > GSH/ASRE > ASRE (Figure 3D).

#### 2.1.3. Effect of ASRE on Na_2_S_4_ Induced Reduction of ^●^cPTIO

Recently, endogenous hydropersulfides and related polysulfides were recognized as important biological mediators [22,23]. Therefore, we studied the ability of the polysulfide Na_2_S_4_/ASRE mixture to reduce ^●^cPTIO. Previously, we have shown that Na_2_S_4_ effectively reduces ^●^cPTIO [28]. The potency of 20 µmol L^−1^ Na_2_S_4_ was comparable to that of ~400 µmol L^−1^ Trolox. The presence of ASRE in the Na_2_S_4_/^●^cPTIO mixture slightly decreased the concentration of ^●^cPTIO when compared to that with the Na_2_S_4_/^●^cPTIO mixture (Figure 4A). However, the dynamics and extent of the ^●^cPTIO decrease in the presence of the Na_2_S_4_/ASRE/^●^cPTIO mixture was less than the cumulative effects of ASRE/^●^cPTIO and Na_2_S_4_/^●^cPTIO (Figure 4B), indicating that ASRE slightly inhibited the potency of Na_2_S_4_ to reduce ^●^cPTIO.

#### 2.1.4. Effect of ASRE on Its Own and in the Mixture with H_2_S, Polysulfides, Cys and GSH on pDNA Cleavage

Next, we examined whether ASRE on its own or in the mixture with H_2_S, polysulfides, Cys or GSH could directly cleave pDNA in vitro, where the contribution of other (unknown) biologically important molecules and/or pathways is eliminated. Briefly, the pDNA cleavage assay can detect any activity that attacks and disrupts the sugar-phosphate backbone of DNA. ASRE alone had a minor but significant cleavage effect on pDNA with a bell-shaped profile. However, in the presence of H_2_S and Cys, and to a minor extent with GSH, pDNA cleavage by ASRE increased in a concentration-dependent manner (Figure 5). H_2_S, Cys or GSH (100 µmol L^−1^) on their own did not cleave pDNA at all. In contrast to these compounds, polysulfides can effectively cleave pDNA in the order Na_2_S_4_ ≥ Na_2_S_3_ > Na_2_S_2_. Importantly, we observed that polysulfide-mediated pDNA cleavage was prevented by ASRE in a concentration-dependent manner (Figure 5).

### 2.2. Experiments Using ASRE Components: CA, CGA, Protocatechuic Acid (PCA), Eleutheroside B (EB) and Eleutheroside E (EE)

#### 2.2.1. CA, CGA and PCA (But Not EB or EE) Interacting with H_2_S Reduce ^●^cPTIO

Next, the potency of five individual components of ASRE, the CA, CGA, PCA, EB and EE, either on their own or in the mixture with H_2_S, to reduce ^●^cPTIO was assessed. UV-VIS spectra of ASRE, CA, CGA, PCA, EB and EE are shown in Figure 6A. CGA alone had a modest radical-reducing effect. However, addition of H_2_S (≥5 µmol L^−1^) to a constant CGA/^●^cPTIO mixture ratio significantly increased the reduction of ^●^cPTIO in an H_2_S concentration-dependent manner (Figure 6B). Furthermore, adding CGA to the constant H_2_S/^●^cPTIO molar ratio increased the reduction of ^●^cPTIO in a concentration-dependent manner. Similar results were observed for CA, which manifested a modest radical-reducing effect (Figure 7A), but an addition of H_2_S (≥5 µmol L^−1^) into a constant CA/^●^cPTIO mixture ratio significantly increased the reduction of ^●^cPTIO in a concentration-dependent manner (Figure 6C). The effect of PCA on ^●^cPTIO reduction was minor and only slightly increased after the addition of H_2_S. EB and EE alone or in a mixture with H_2_S had no effect on ^●^cPTIO concentration (Figure 6C). The order of potency to reduce ^●^cPTIO was CA ≥ CGA > PCA ≥ EB = EE = 0 (Figure 7A). Potency to reduce ^●^cPTIO in a mixture with H_2_S was as follows: CA ~ CGA > PCA > EB = EE = 0 (Figure 6C).

#### 2.2.2. CA, CGA, PCA, EB or EE Interacting with Polysulfide Do Not Reduce ^●^cPTIO

Polysulfide Na_2_S_4_ alone effectively reduced ^●^cPTIO (Figure 4A and Figure 7B). The presence of ASRE components, CA, CGA, PCA, EB and EE in the ^●^cPTIO/Na_2_S_4_ mixture slightly decreased the ^●^cPTIO concentration. However, the rate and extent of the ^●^cPTIO decrease in the presence of Na_2_S_4_/ASRE component/^●^cPTIO mixture was similar to the cumulative effects of the ASRE component/^●^cPTIO and Na_2_S_4_/^●^cPTIO (Figure 7B,C). Data for EB and EE are not shown. This indicates that ASRE components do not influence the potency of Na_2_S_4_ to reduce ^●^cPTIO.

#### 2.2.3. CA, CGA and PCA (But Not EB or EE) in the Mixture with Cys or GSH Reduce ^●^cPTIO

In addition, we compared the potency of five selected individual components of ASRE to reduce ^●^cPTIO after their interaction with Cys (Figure 8A) and GSH (Figure 8B). The order of their potency to reduce ^●^cPTIO in the mixture with Cys was: CA > CGA > PCA > EB ~ EE ~ Cys = 0. The order of their potency to reduce ^●^cPTIO in the mixture with GSH was: CA ~ CGA > PCA > EB ~ EE ~ GSH = 0. When the data on the potency of the ASRE components mixed with H_2_S (Figure 6) were included into the comparison, the order of ^●^cPTIO-reducing potency was as follows: ASRE component/H_2_S > ASRE component/Cys > ASRE component/GSH.

#### 2.2.4. Effect of the ASRE Components on pDNA Cleavage

Because of its low solubility in the buffer, EE was not included in the pDNA cleavage experiments. H_2_S (100 µmol L^−1^) alone or mixed with the increased concentrations of CA, CGA, PCA or EB, did not induce cleavage of pDNA. Polysulfides (100 µmol L^−1^) induced pDNA cleavage in the order of potency Na_2_S_4_ > Na_2_S_2_, which is in accord with Figure 5. Notably, CA, CGA or PCA inhibited polysulfide-induced pDNA cleavage in a concentration-dependent manner (Figure 9). However, in the presence of polysulfides, EB slightly increased pDNA cleavage at lower concentrations and decreased it at higher concentrations (Figure 9D).

## 3. Discussion

AS and its phytochemical individual components possess numerous diverse pharmacological effects, including antioxidant properties, yet their molecular mechanism is not fully understood [2]. Herein, we provide evidence that the interaction of ASRE and some of its individual components with H_2_S, Cys and GSH, but not with Na_2_S_4_, GSSG, MET or cystine, significantly increased its antioxidant potency to the level comparable or even more effective than that of Trolox. Similarly, an increased antioxidant potency has been observed after the interaction of H_2_S with NO, selenite, phthalic selenoanhydride or doxycycline [28,29,30,31]. However, detailed information on how the biologically active compounds stimulate the antioxidant potency of ASRE is not available yet.

H_2_S is produced endogenously and exerts relevant biological effects and functions. The H_2_S concentration in plasma is less than 1 µmol L^−1^. However, in local microenvironments where it is enzymatically produced, it can reach higher levels [32]. Therefore, the physiological significance of the H_2_S/ASRE antioxidant effects will need to be determined.

The concentration of total Cys in the serum/plasma is ~250 μmol L^−1^ [33]. GSH is the most abundant nonprotein thiol in mammalian cells, reaching an intracellular concentration in the mmol L^−1^ range, whereas its plasma concentration does not exceed micromolar range [34]. The increased antioxidant effect of the Cys/ASRE and GSH/ASRE mixture found in vitro are within the physiological concentrations of Cys and GSH and can thus be physiologically relevant in the thiols/ASRE interactions in a living organism.

Whereas the H_2_S/ASRE interaction increased the reduction of the ^●^cPTIO radical, the Na_2_S_4_/ASRE interaction had an opposite minor effect: it inhibited the reduction of ^●^cPTIO induced by Na_2_S_4_. Since Na_2_S_4_ alone possesses high potency to reduce ^●^cPTIO, we assume that the inhibitory effect of ASRE may result from “scavenging” of Na_2_S_4_ leading to a decrease of Na_2_S_4_ redox capacity in the system.

In the pDNA cleavage assay, high concentrations of the compounds were used in order to detect any cleavage. However, it should be noticed that even tiny DNA damage levels, not detectable by our pDNA cleavage assay, may have significant physiological consequences. Minor pDNA cleavage caused by ASRE on its own may indicate that it contains active unknown species responsible for the effects. The molecular mechanism of the bell-shaped increase of pDNA cleavage by interaction of ASRE with H_2_S or Cys is unknown. Similarly, the physiological significance of pDNA cleavage in vitro in a complex biological system, at in situ concentrations of ASRE, is unknown.

Na_2_S_n_ induced pDNA cleavage in the same manner as in our previous studies [28,31]. ASRE and its components, CA, CGA and PCA, but not EB, decreased the concentration of ^●^cPTIO. Similarly, ASRE, CA, CGA and PCA, but not EB at a lower concentration, inhibited polysulfide-induced pDNA cleavage. If we assume that the pDNA cleavage reaction requires the action of certain radicals, we may suggest that the inhibitory effect of ASRE, CA, CGA, PCA, and to less extend of EB, on pDNA cleavage can result from the “scavenging” of the radicals by these compounds.

It is of interest to know which particular ASRE components are responsible for the ASRE antioxidant effects. ASRE and several dozen compounds isolated from ASRE possess numerous biological effects, including antioxidant properties. Dried root ethanolic extract from AS has been found to scavenge superoxide anion and hydroxyl radicals [4]. CA and CGA isolated from AS have antioxidant activities [5], whereas EB and EE have no significant antioxidant (anti-DPPH) properties [6,8]. To extend these observations, CA, CGA, PCA, EB and EE as ASRE components [2] were selected for our study. We confirmed some of the previous observations, but we also found the important fact that the antioxidant properties of CA and CGA were strongly enhanced after interaction with H_2_S, Cys or GSH. This increase is similar to that observed for ASRE, implying that these two agents (and possibly others that were not studied) are responsible for increasing their antioxidant properties and inhibiting polysulfide-induced pDNA cleavage. PCA had only a partial effect compared to that of CA or CGA, while EB and EE had no effect at all. CA, CGA and to a lesser extent PCA interacted with H_2_S, Cys and GSH but not with GSSG, MET or cystine, indicating that the presence of thiol groups was a prerequisite for the observed effects.

The concentrations of H_2_S, Cys and GSH differ in different parts of a living organism and can change in different pathological conditions [32,33,34]. From our results, we hypothesize that the antioxidant properties of ASRE and its components depend on the concentrations of H_2_S, Cys, and GSH in situ. Two central thiol/disulfide redox couples in human plasma, Cys/cystine and GSH/GSSG, vary little among healthy individuals [35]. However, changes of the thiol/disulfide ratio regulate and are implicated in many biological processes and diseases, including enzyme catalysis, gene expression, and pathway signaling [36,37]. If the increase of the antioxidant properties of the ASRE/thiol mixture found in vitro takes place in a living organism, then changes of the thiol/disulfide ratio can influence the biological effects of ASRE and its components.

In conclusion, we found that H_2_S/HS^−^, Cys and GSH interacting with ASRE, CA and CGA increased their antioxidant properties and modulated cleavage of plasmid DNA. These findings can contribute to understanding many biological effects of ASRE, especially under conditions where radicals play a significant role, and suggest that antioxidant properties of ASRE and its components depend on the concentrations of H_2_S, Cys and GSH, and changes of the thiol/disulfide ratio in situ. We may assume that the interaction of H_2_S/HS^−^, Cys and GSH with ASRE, CA and CGA can contribute to their numerous biological effects. However, whether and to what extent the in vitro data on ^●^cPTIO reduction in a phosphate buffer and pDNA cleavage can be directly implicated in the complex biological environment requires further study.

## 4. Materials and Methods

### 4.1. Chemicals and Solutions

The following chemicals were purchased from Sigma-Aldrich (Schnelldorf, Germany): 2-(4-carboxyphenyl)-4,5-dihydro-4,4,5,5-tetramethyl-1H-imidazol-1-yloxy-3-oxide potassium salt (^●^cPTIO, C221), (±)-6-hydroxy-2,5,7,8-tetramethylchromane-2-carboxylic acid (Trolox; 238813), caffeic acid (CA, C0625, purity ≥ 98.0%), eleutheroside B (EB, 90974, purity ≥ 98.0%), eleutheroside E (EE, 08198, purity ≥ 98.0%), L-methionine (MET, M9625), L-cysteine hydrochloride (Cys, C1276), L-glutathione reduced (GSH, G4251), glutathione oxidized (GSSG, G-6654), L-cystine dihydrochloride (cystine, C-6727), diethylenetriaminepentaacetic acid (DTPA, D6518), sodium phosphate monobasic (NaH_2_PO_4_, S5011) and sodium phosphate dibasic (Na_2_HPO_4_, S7907). Chlorogenic acid (CGA, PHR2202, purity 97.2%) was from Supelco (Schnelldorf, Germany), and protocatechuic acid (PCA, PHL89766, purity ≥ 98.0%) from Merck (Taufkirchen, Germany). Stock solutions (10 mmol L^−1^) of Cys and GSH were prepared just before application. Na_2_S (100 mmol L^−1^), polysulfides Na_2_S_2_, Na_2_S_3_ and Na_2_S_4_ (20 mmol L^−1^) were prepared in argon-bubbled deionised distilled H_2_O, immediately frozen at −80 °C and used just after thawing.

### 4.2. Preparation of Aqueous Root Extract from Acanthopanax Senticosus (ASRE)

Dried roots of *Acanthopanax senticosus* (AS, Siberian ginseng) were obtained from the arboretum in Liptovsky Hradok (Slovakia) in November 2021. The roots were crushed in liquid nitrogen to obtain fine pieces. Five hundred milligrams of these pieces were incubated in 10 mL of the buffer consisting of 0.9% NaCl and 10 mmol L^−1^ sodium phosphate (pH 7.0) at 80 °C for 120 min. After incubation and cooling of the solution, the upper clear ASRE extract was collected (without a very top layer, where light insoluble particles floated). The extract was divided into 400 µL aliquots and stored at –20 °C for several days. It contained ~7.5 mg dry matter *per* mL. UV-VIS spectra of ASRE were used to adjust its concentration in experiments from three different ASRE preparations.

### 4.3. UV-VIS of ^●^cPTIO Radical

To obtain 1 mL of the working solution, 1–50 µL of stock solution of the studied compounds was added to the appropriate volume (950–999 µL) of 100 mmol L^−1^ sodium phosphate buffer (pH 7.4, 37 °C) containing the final concentrations of 100 µmol L^−1 ●^cPTIO and 100 µmol L^−1^ DTPA. UV-VIS absorption spectra (900–200 nm) were recorded every 30 s for 30 min with a Shimadzu 1800 (Kyoto, Japan) spectrometer at 37 °C. In all experiments, the absorbance (ABS) path length of 10 mm was used. Reduction of the ^●^cPTIO radical was determined as the decrease of the absorbance at 560 nm [29,38]. Concentration and stability of the studied compounds were checked by their UV-VIS spectra at the beginning of particular experiment in the range of 200–900 nm.

### 4.4. Plasmid DNA Cleavage

A pDNA cleavage assay with the use of pBR322 plasmid (New England BioLabs, Inc., N3033 L, Ipswich, MA, USA) was performed as reported previously [31]. In this assay, all samples contained 0.2 μg pDNA in the final sodium phosphate buffer (25 mmol L^−1^ sodium phosphate, 50 μmol L^−1^ DTPA, pH 7.4). Five microliters of the Na_2_S, Na_2_S_n_, Cys or GSH aqueous stock solution (in final 100 μmol L^−1^) was added to 15 µL of pDNA solution containing ASRE (in mg mL^−1^: 0, 0.23, 0.47, 0.94, 1.41 or 1.88) or its chemical component (in mmol L^−1^: 0.2, 0.4, 0.8, 1.2 or 1.6). All listed concentrations were final and calculated for a 20 µL sample. All ASRE components were prepared in water, except PA, which was dissolved in 100 mmol L^−1^ sodium phosphate. The resulting samples were incubated for 30 min at 37 °C. After incubation, the reaction mixtures were subjected to 0.6% agarose gel electrophoresis. Integrated densities of all pBR322 forms in each lane were quantified using Image Studio analysis software (LI-COR Biotechnology, Bad Homburg, Germany) to estimate pDNA cleavage efficiency.

## Figures and Tables

**Figure 1 molecules-27-05735-f001:**
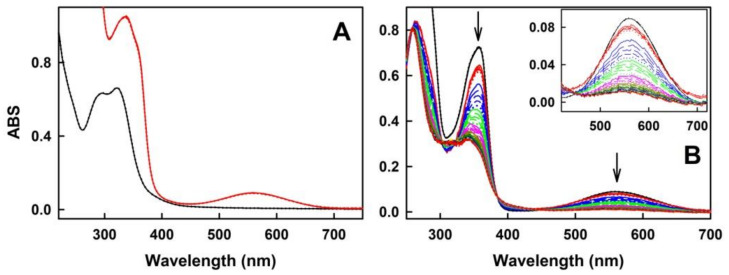
Representative time-resolved UV-VIS spectra (absorbance, ABS) of aqueous root extract from *Acanthopanax senticosus* (ASRE) without and with 2-(4-carboxyphenyl)-4,5-dihydro-4,4,5,5-tetramethyl-1H-imidazol-1-yloxy-3-oxide radical (^●^cPTIO) and Na_2_S. (**A**) UV-VIS spectra of ASRE (75 µg mL^−1^) without (black) and with (red) ^●^cPTIO (100 µmol L^−1^) measured every 30 s for 2.5 min in 100 mmol L^−1^ sodium phosphate, 100 µmol L^−1^ diethylenetriaminepentaacetic acid (DTPA), pH 7.4 buffer at 37 °C. (**B**) Representative time-resolved UV-VIS spectra of 100 µmol L^−1 ●^cPTIO (black) and its mixture with 150 µg mL^−1^ of ASRE (red) after the addition of 25 µmol L^−1^ Na_2_S measured every 30 s for 30 min. The absorbance (ABS) at 358 and 560 nm decreased over time (arrows). The background ASRE spectrum was subtracted. The time-resolved spectra have this sequence: the black solid line is followed each 30 s by: long dash black, medium dash black, short dash black, dotted black, solid red line, long dash red, medium dash red, etc. Insert: Details of ABS at 420–720 nm.

**Figure 2 molecules-27-05735-f002:**
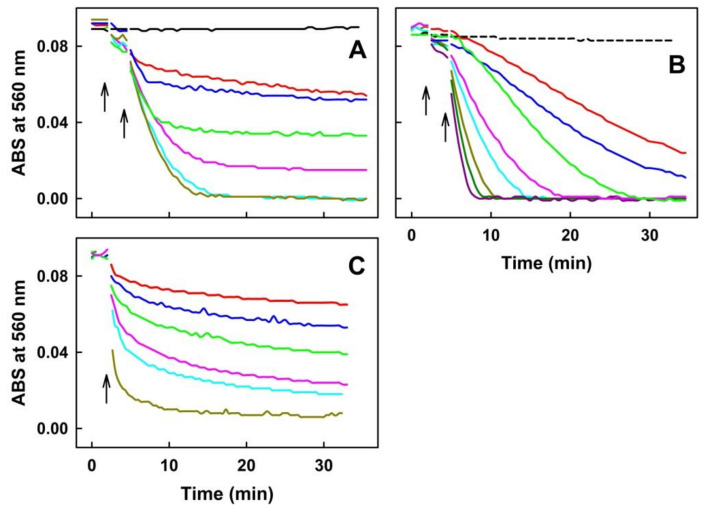
The effects of the ASRE/Na_2_S mixture and Trolox on the reduction of ^●^cPTIO measured as a decrease of ABS at 560 nm. (**A**) Time dependence of ABS at 560 nm of ^●^cPTIO (100 µmol L^−1^) alone (black) and in the ASRE/^●^cPTIO (150 µg mL^−1^/100 µmol L^−1^) mixture with increasing concentrations of Na_2_S: 0 (red), 5 (blue), 10 (green), 20 (pink), 50 (cyan) and 100 (dark yellow) µmol L^−1^ Na_2_S. (**B**) Time dependence of ABS at 560 nm of the Na_2_S/^●^cPTIO (400/100 in µmol L^−1^) mixture (dash black) with increasing concentrations of ASRE: 6 (red), 12 (blue), 24 (green), 37.5 (pink), 75 (cyan), 150 (dark yellow), 225 (dark green) and 300 µg mL^−1^ (dark red) ASRE. Addition of ASRE to ^●^cPTIO is marked by the left arrow, and subsequent addition of Na_2_S by the right arrow. (**C**) Time dependence of ABS at 560 nm of ^●^cPTIO (100 µmol L^−1^) with the increasing concentration of (±)-6-hydroxy-2,5,7,8-tetramethylchromane-2-carboxylic acid (Trolox): 50 (red), 100 (blue), 200 (green), 400 (pink), 500 (cyan) and 2000 (dark yellow) µmol L^−1^ Trolox. The arrow marks addition of Trolox to the ^●^cPTIO solution. Buffer: 100 mmol L^−1^ sodium phosphate, 100 µmol L^−1^ DTPA, pH 7.4, measured at 37 °C.

**Figure 3 molecules-27-05735-f003:**
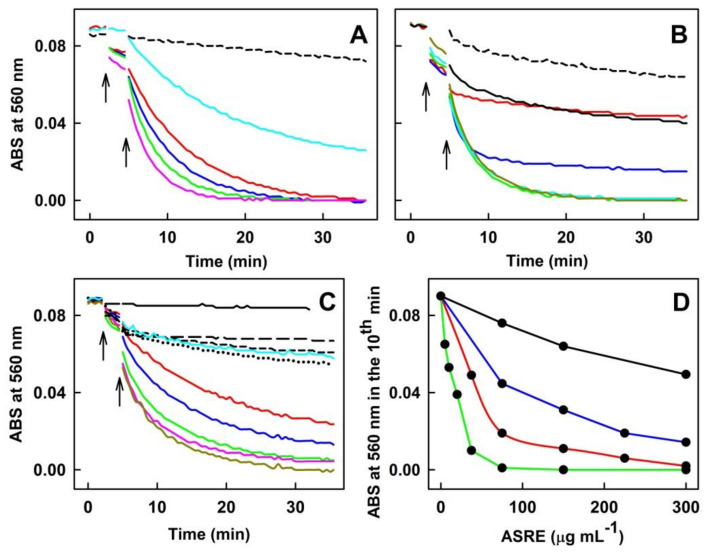
The effect of ASRE/Cys, ASRE/GSH, ASRE/GSSG, ASRE/MET and ASRE/cystine mixtures on the reduction of the ^●^cPTIO, as measured by the decrease of ABS at 560 nm. (**A**) Time dependence of ABS of Cys with the ^●^cPTIO (400/100 in µmol L^−1^) mixture (dash black) in the presence of 37.5 (cyan), 75 (red), 150 (blue), 225 (green) and 300 (pink) µg mL^−1^ of ASRE. UV-VIS spectra of ^●^cPTIO were measured every 30 s for 2.5 min, followed by addition of ASRE (left arrow) and measurement for 2.5 min, then, Cys was added (right arrow) and measured every 30 s for 30 min. (**B**) Time dependence of ABS of ^●^cPTIO (100 µmol L^−1^) with 150 (dash black) or 300 µg mL^−1^ (black) ASRE. Time dependence of the mixture of ASRE (300 µg mL^−1^) with ^●^cPTIO (100 µmol L^−1^) in the presence of 2 (red), 10 (blue), 25 (green), 100 (cyan) and 200 (dark yellow) µmol L^−1^ Cys. UV-VIS spectra of ^●^cPTIO were measured every 30 s for 2.5 min, followed by addition of ASRE (left arrow) and measurement for 2.5 min, then, Cys was added (right arrow) and measured every 30 s for 30 min. (**C**) Time dependence of ABS of the mixture of ASRE (150 µg mL^−1^) with ^●^cPTIO (100 µmol L^−1^) (cyan) and GSH. ABS at 560 nm of the GSH/^●^cPTIO (400/100 in µmol L^−1^) mixture without (black) and with 75 (red), 150 (blue), 225 (green) and 300 (pink) µg mL^−1^ of ASRE. The mixture of 300 µg mL^−1^ ASRE with GSH/^●^cPTIO (2000/100 in µmol L^−1^) (dark yellow). Time dependence of ABS of the mixture of ASRE (150 µg mL^−1^) with ^●^cPTIO (100 µmol L^−1^) in the presence of 400 µmol L^−1^ of GSSG (long dash black), MET (short dash black) or cystine (dotted black). UV-VIS spectra of ^●^cPTIO were measured every 30 s for 2.5 min, followed by addition of ASRE (left arrow) and measured for 2.5 min, then GSH, GSSG, MET and cystine were added (right arrow) and measured every 30 s for 30 min. (**D**) The effect of increased concentration of ASRE without (black) and with H_2_S (green), Cys (red) and GSH (blue) on the reduction of ^●^cPTIO (100 µmol L^−1^). ABS at 560 nm was measured in the 10th min after compound addition.

**Figure 4 molecules-27-05735-f004:**
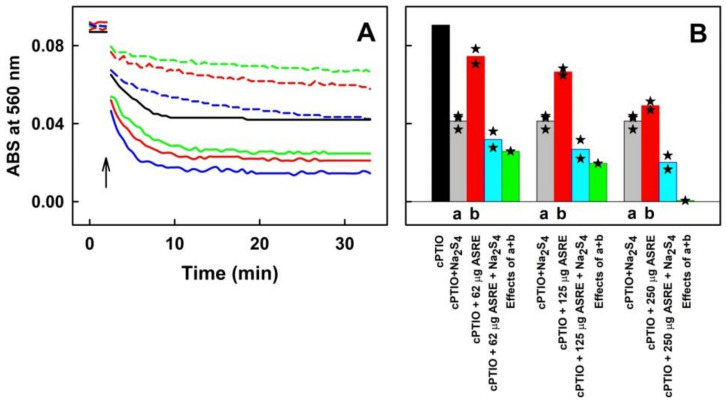
The effects of the Na_2_S_4_/ASRE mixture on ^●^cPTIO reduction. (**A**) Representative time dependence of ABS of ^●^cPTIO (100 µmol L^−1^) with 62 (dash green), 125 (dash red) and 250 (dash blue) µg mL^−1^ of ASRE. Time dependence of ABS of Na_2_S_4_/^●^cPTIO (20/100 in µmol L^−1^) at 560 nm (black) and after addition of 62 (green), 125 (red) and 250 (blue) µg mL^−1^ of ASRE (marked by arrow). (**B**) Comparison of the average ABS values at 560 nm in the tenth min of ^●^cPTIO (100 µmol L^−1^) (black), Na_2_S_4_/^●^cPTIO (20/100 in µmol L^−1^) mixture (gray), ^●^cPTIO (100 µmol L^−1^) with 62, 125 and 250 µg mL^−1^ of ASRE (red) and Na_2_S_4_/^●^cPTIO (20/100 in µmol L^−1^) with 62, 125 and 250 µg mL^−1^ of ASRE (cyan). Green bars represent the calculated average sum effects of Na_2_S_4_/^●^cPTIO (20/100 in µmol L^−1^, gray) and cPTIO with 62, 125 and 250 µg mL^−1^ of ASRE (red). Stars indicate individual experiments (*n* = 2 or 3).

**Figure 5 molecules-27-05735-f005:**
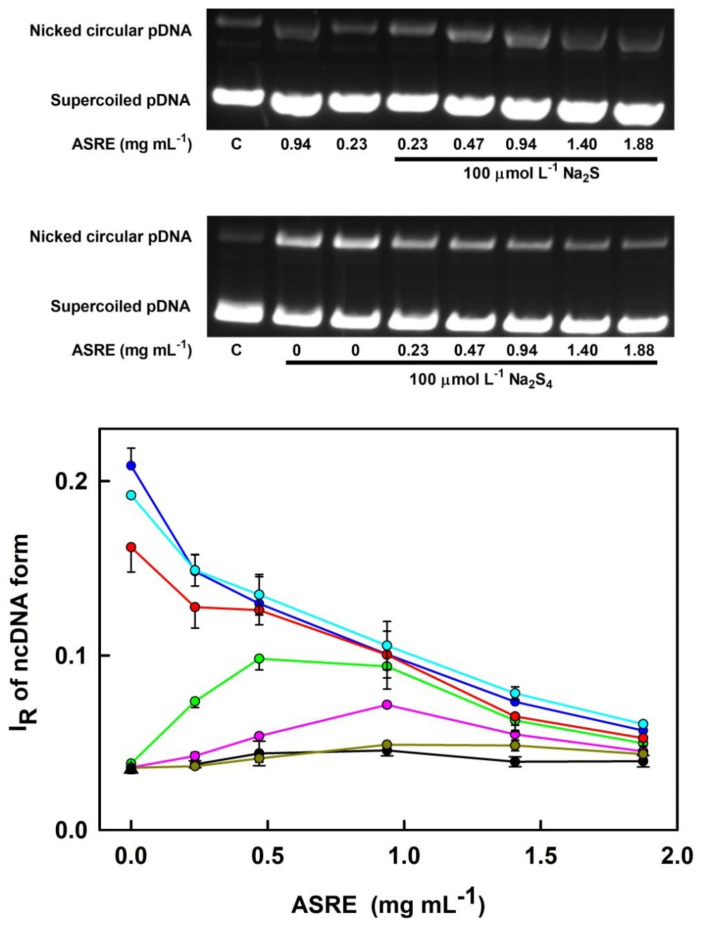
The effect of ASRE on pDNA cleavage in the presence of Na_2_S, polysulfides, GSH and Cys. The upper figure shows representative agarose gels indicating pDNA cleavage in the presence of an increasing concentration of ASRE and 100 μmol L^−1^ Na_2_S or Na_2_S_4_. The band at the bottom corresponds to the circular supercoiled form of pDNA, and the top band corresponds to the nicked circular form of pDNA. The letter C indicates control pDNA without any treatment. The final concentration of pDNA was 0.2 μg in 20 μL. The lower figure presents concentration-dependent effects of ASRE on Na_2_S-, polysulfide-, GSH- and Cys-mediated cleavage of pDNA in the 25 mmol L^−1^ sodium phosphate buffer (50 μmol L^−1^ DTPA, pH 7.4) at 37 °C. Control pDNA (triangle). Concentration-dependent effect of ASRE without (black) and with 100 μmol L^−1^ GSH (dark yellow), Cys (pink), Na_2_S (green), Na_2_S_2_ (red), Na_2_S_3_ (cyan) and Na_2_S_4_ (blue). The final concentration of pDNA was 0.2 μg in 20 μL. I_R_ of ncDNA form represents the relative intensity of the nicked circular pDNA. Horizontal black marks indicate the means ± SD, *n* = 3–6. Statistical significance of the effects of ASRE alone was evaluated by one-way ANOVA (*p* < 0.0001) followed by Dunnett’s test with a significant difference for 0.47 (*p* < 0.01), 0.94 (*p* < 0.0001) and 1.41 (*p* < 0.01) mg mL^−1^ ASRE.

**Figure 6 molecules-27-05735-f006:**
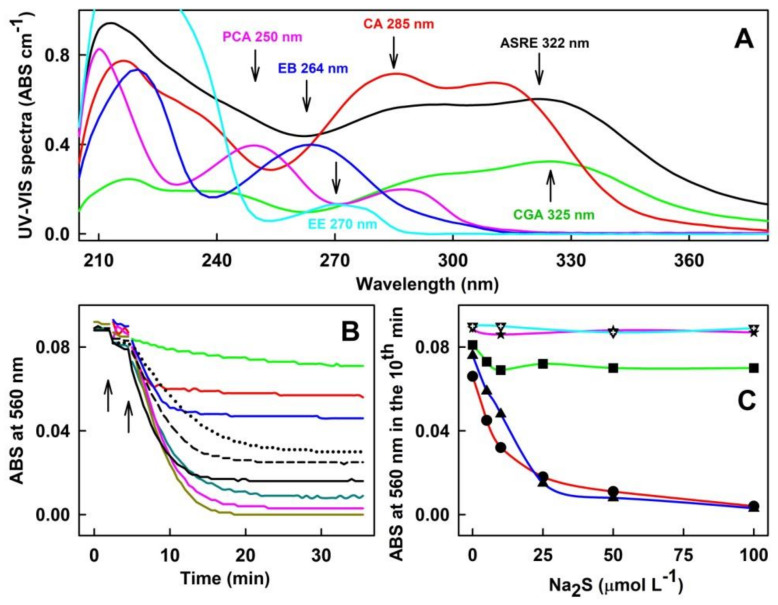
The effects of ASRE components on the reduction of ^●^cPTIO measured as a decrease of ABS at 560 nm. (**A**) Representative UV-VIS spectra of 75 µg mL^−1^ ASRE (black; peak at 322 nm), 20 µmol L^−1^ CGA (green; peak at 325 nm), 50 µmol L^−1^ CA (red; peak at 285 nm), 50 µmol L^−1^ PCA (pink; peak at 250 nm), 25 µmol L^−1^ EB (blue; peaks at 264 and 220 nm) and 50 µmol L^−1^ EE (cyan; peak at 270 nm). Buffer: 100 mmol L^−1^ sodium phosphate, 100 µmol L^−1^ DTPA, pH 7.4, 37 °C. (**B**) Time dependence of ABS at 560 nm of the Na_2_S/CGA/^●^cPTIO mixture. ABS of the CGA/^●^cPTIO (50/100 in µmol L^−1^) mixture without (green) and with 5 (red), 10 (blue), 20 (dash black), 25 (dark cyan), 50 (pink) and 100 (dark yellow) µmol L^−1^ Na_2_S. The addition of CGA into ^●^cPTIO is marked by the left arrow, and the subsequent addition of Na_2_S by the right arrow. ABS of the Na_2_S/^●^cPTIO (20/100 in µmol L^−1^) mixture with 20 (dotted black), 50 (dash black) and 100 (black) µmol L^−1^ CGA. Buffer: 100 mmol L^−1^ sodium phosphate, 100 µmol L^−1^ DTPA, pH 7.4, 37 °C. (**C**) The ABS at 560 nm in the tenth min after 50 µmol L^−1^ addition of CA (red, circle), CGA (blue, triangle), PCA (green, square), EB (pink, star) and EE (cyan, open triangle) to the ^●^cPTIO (100 µmol L^−1^) containing increased concentration of Na_2_S. Buffer: 100 mmol L^−1^ sodium phosphate, 100 µmol L^−1^ DTPA, pH 7.4, 37 °C.

**Figure 7 molecules-27-05735-f007:**
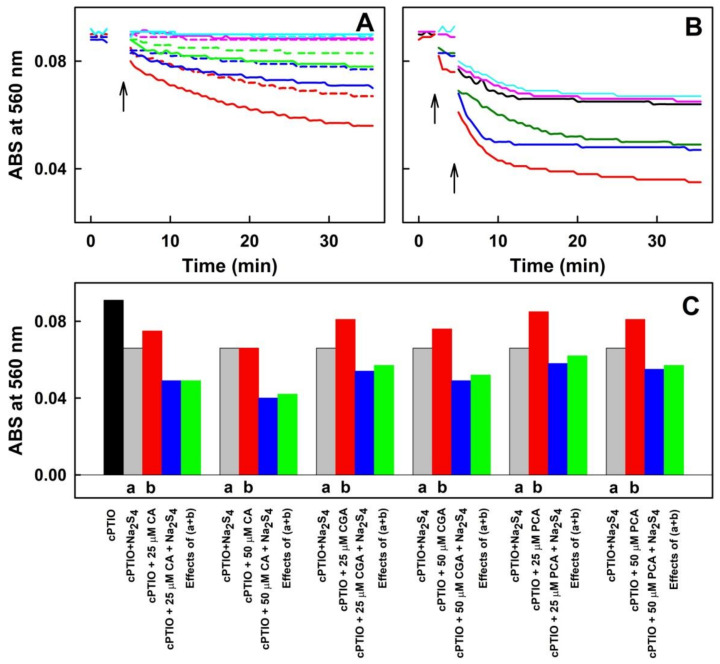
The effect of ASRE components CA, CGA and PCA in a mixture with Na_2_S_4_ on the reduction of ^●^cPTIO. (**A**) Representative time dependence of ABS at 560 nm of ^●^cPTIO (100 µmol L^−1^) with 25 (dash lines) and 50 (full lines) µmol L^−1^ of CA (red), CGA (blue), PCA (green), EB (pink) and EE (cyan). Addition of compounds is marked by arrow. (**B**) Representative time dependence of ABS of Na_2_S_4_/^●^cPTIO (10/100 in µmol L^−1^) at 560 nm (black) and after addition of 50 µmol L^−1^ of CA (red), CGA (blue), PCA (green), EB (pink) and EE (cyan). Addition of Na_2_S_4_ is marked by the left arrow and addition of compounds by the right arrow. (**C**) Comparison of ABS values at 560 nm in the tenth minutes after application of 100 µmol L^−1 ●^cPTIO alone (black) and with 10 µmol L^−1^ Na_2_S_4_ (gray), 25 or 50 µmol L^−1^ ASRE components (red) and with the ASRE component/Na_2_S_4_ (25/10 or 50/10 in µmol L^−1^) mixture (blue). Green bars represent the calculated average sum effects of Na_2_S_4_/^●^cPTIO (10/100 in µmol L^−1^, gray) and ^●^cPTIO with CA, CGA and PCA (25 or 50 µmol L^−1^, red).

**Figure 8 molecules-27-05735-f008:**
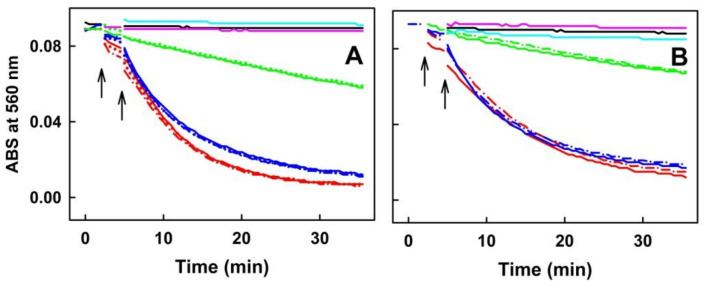
Time dependence of ABS at 560 nm of ^●^cPTIO (100 µmol L^−1^) in the presence of the ASRE component with Cys or GSH. (**A**) ABS of the Cys/^●^cPTIO (50/100 in µmol L^−1^, black) mixture with 50 µmol L^−1^ of CA (red, dash-dot, dotted), CGA (blue, dash-dot, dotted), PCA (green, dotted), EB (pink) and EE (cyan). (**B**) ABS of the GSH/^●^cPTIO (50/100 in µmol L^−1^, black) mixture with 50 µmol L^−1^ of CA (red, dash-dot), CGA (blue, dash-dot), PCA (green, dash-dot), EB (pink) and EE (cyan); *n* = 2 or 3. The addition of the compound to ^●^cPTIO is marked by the left arrow, and the subsequent addition of Cys or GSH by the right arrow. Buffer: 100 mmol L^−1^ sodium phosphate, 100 µmol L^−1^ DTPA, pH 7.4, 37 °C.

**Figure 9 molecules-27-05735-f009:**
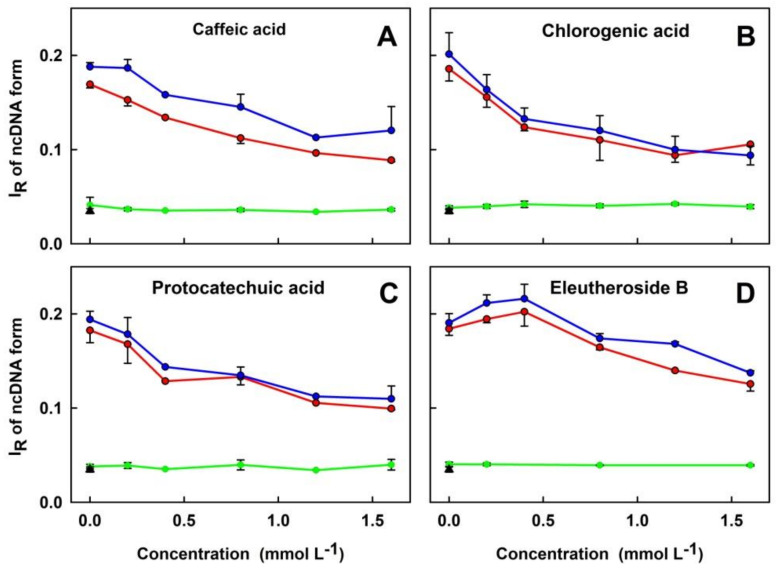
The concentration-dependent effects of the ASRE components on polysulfide-induced pDNA cleavage in the 25 mmol L^−1^ sodium phosphate buffer, 50 μmol L^−1^ DTPA, pH 7.4, 37 °C. Control pDNA without any treatment (triangle). The concentration-dependent effects of CA (**A**), CGA (**B**), PCA (**C**) and EB (**D**) in the presence of 100 μmol L^−1^ Na_2_S (green), Na_2_S_2_ (red) or Na_2_S_4_ (blue). I_R_ of ncDNA form represents the relative intensity of the nicked circular pDNA. Data represent average of 3–6 values for concentrations of 0.2, 0.8 and 1.6 mmol L^−1^ and 1–3 values for 0.4 and 1.2 mmol L^−1^ concentrations. Horizontal black marks indicate means ± SD.

## Data Availability

The data presented in this study are available on request from the corresponding author.

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
