# Peer review of "Extract of Acanthopanax senticosus and Its Components Interacting with Sulfide, Cysteine and Glutathione Increase Their Antioxidant Potencies and Inhibit Polysulfide-Induced Cleavage of Plasmid DNA"

_molecules, 2022, doi:10.3390/molecules27175735_

Round 1

Reviewer 1 Report

I. 33-34:Please do not use the word herb keeping in mind the shrub. The correct plant description is: Acanthopanax senticosus (Rupr. et Maxim.) Maxim. or Eleutherococcus senticosus (Rupr. et Maxim.) Maxim.

II. - 4.1. Please provide information about a purity of plant-based standards (99.9%?)

III. 4.2. Why did you not concentrate the extract to obtain a solid extract (dried extract) and then resolved in a known amount of water of buffer?

IV. Please add (emphasize) the conclussion part to be more vissiable in the text.

Author Response

Responses to the Reviewer 1. Manuscript ID: molecules-1872169

Thank you for your valuable remarks.

General comment 1.

English language and style are fine/minor spell check required
Answer 1.

We improved the English language style and checked the spelling.

General comment 2.

Are the conclusions supported by the results? Must be improved.

Answer 2.

In the revised manuscript: We improved the conclusions supported by the results.

Comment 1.1.

  1. 33-34:Please do not use the word herb keeping in mind the shrub. The correct plant description is: Acanthopanax senticosus (Rupr. et Maxim.) Maxim. or Eleutherococcus senticosus (Rupr. et Maxim.) Maxim.

Answer 1.1.

In the revised manuscript: We do not use the word herb and used the correct plant description: Acanthopanax senticosus (Rupr. et Maxim.) Maxim. 4.1.

Comment 1.2.

  1. - 4.1. Please provide information about a purity of plant-based standards (99.9%?)

Answer 1.2.

In the revised manuscript: In the section “Chemicals and Methods” we included the purity of the plant-based standards.

Comment 1.3.

III. 4.2. Why did you not concentrate the extract to obtain a solid extract (dried extract) and then resolved in a known amount of water of buffer?

Answer 1.3.

Since we observed that the concentration of the extract in the buffer, obtained by described procedure, was right suitable for our study, we did not concentrate the extract to obtain a solid extract (dried extract) and then resolved in a known amount of the buffer.

In our case the concentration was checked by the absorbance of UV-Vis spectra of the extract, and it was similar in three extractions within < ± 10%. The concentration was adjusted by the addition of the buffer or by volume of the extract used in the experiments. We concentrated the extract to obtain a solid extract (dried extract) in order to found concentration of the matter in extracted solution only. In our procedure, we bypassed the possibility that some chemicals in the extract are unstable and/or could react with each other when the solution is evaporated, where their concentration will be extremely increased.

Comment 1.4.

  1. Please add (emphasize) the conclusion part to be more visible in the text.

Answer 1.4.

We emphasized the conclusion part in “Conclusion.” to be more visible in the text.

Reviewer 2 Report

1.      Pay attention to abbreviations (first mention - define it, not later, and stick to it consistently throughout the text).

2.      There is a problem of spacing in between the words, make it correct.

3.      Manuscript draft is incomplete.

4.      No new data is available in the manuscript.

5.      Already lots of work have been published in the concern topic.

Author Response

Responses to the Reviewer 2. Manuscript ID: molecules-1872169

Thank you for your valuable remarks.

General comment 1.

Moderate English changes required

Answer 1.

To our best, we improved the English language.

General comment 2.

Does the introduction provide sufficient background and include all relevant references? Must be improved.

Answer 2.

In the revised manuscript: In introduction we provide more sufficient background included more references relevant to the research.

General comment 3.

Are all the cited references relevant to the research? Must be improved.

Answer 3.

In the revised manuscript: We included more references relevant to the research.

General comment 4.

Are the conclusions supported by the results? Must be improved.

Answer 4.

In the revised manuscript: We improved the conclusions supported by the results, see “Discussion”.

Comment 2.1.

  1. Pay attention to abbreviations (first mention - define it, not later, and stick to it consistently throughout the text).

Answer 2.1.

In the revised manuscript: We pay attention to abbreviations, however for practical reason full names were used in the section “4.1. Chemicals and Solutions”.

Comment 2.2.

  1. There is a problem of spacing in between the words, make it correct.

Answer 2.2.

The spacing in between the words depends on provided template.

Comment 2.3.

  1. Manuscript draft is incomplete.

Answer 2.3.

We expanded the content of the revised manuscript.

Comment 2.4.

  1. No new data is available in the manuscript.

Answer 2.4.

We do not understand your comment!

To our knowledge, all the results described in our manuscript regarding the antioxidant properties of the extract and its components, based on their interactions with H2S, cysteine and glutathione and with plasmid DNA, are completely new.

Particularly:

1.) H2S (which is present in living organisms) interacted with the extract and thus significantly increased its antioxidant properties similar or even higher than vitamin E derivative trolox.

2.) Cysteine (which is present in living organisms) interacted with the extract and thus significantly increased its antioxidant properties similar or even higher than vitamin E derivative trolox.

3.) Glutathione (which is present in living organisms) interacted with the extract and thus significantly increased its antioxidant properties similar or even higher than vitamin E derivative trolox.

4.) On the other hand, polysulfides, oxidized glutathione, cystine or methionine (which are present in living organisms) did not increased its antioxidant properties.

5.) H2S and cysteine (which are present in living organisms) interacted with the extract and thus cleavage plasmid DNA in bell-shape concentration dependencies.

6.) Extract, in the concentration dependent manner, inhibited DNA damage caused by polysulfides.

7.) Looking for the components of the extract, which are responsible for the above described effects, we found that the extract components: Caffeic acid, chlorogenic acid and to less extent protocatechuic acid interacted with H2S and thus significantly increased their antioxidant properties.

8.) The extract components: Caffeic acid, chlorogenic acid and to less extent protocatechuic acid interacted with cysteine and thus significantly increased their antioxidant properties.

9.) The extract components: Caffeic acid, chlorogenic acid and to less extent protocatechuic acid interacted with glutathione and thus significantly increased their antioxidant properties.

10.) Possible interaction of the extract components eleutheroside B or eleutheroside E with H2S, cysteine or glutathione did not increased their antioxidant properties.  

11.) Caffeic acid, chlorogenic acid, protocatechuic acid and eleutheroside B (at high concentrations) inhibited DNA damage caused by polysulfides.

 Comment 2.5.

  1. Already lots of works have been published in the concern topic.

Answer 2.5.

We do not understand your comment!

To our knowledge, nothing was published concerning the topic of the antioxidant properties of the extract and its components, based on their interaction with H2S, cysteine and glutathione and with plasmid DNA.

Particularly:

1.) To our knowledge, nothing was published about that: H2S (which is present in living organisms) interacted with the extract and thus significantly increased its antioxidant properties similar or even higher than vitamin E derivative trolox.

2.) To our knowledge, nothing was published about that: Cysteine (which is present in living organisms) interacted with the extract and thus significantly increased its antioxidant properties similar or even higher than vitamin E derivative trolox.

3.) To our knowledge, nothing was published about that: Glutathione (which is present in living organisms) interacted with the extract and thus significantly increased its antioxidant properties similar or even higher than vitamin E derivative trolox.

4.) To our knowledge, nothing was published about that: On the other hand, polysulfides, oxidized glutathione, cystine or methionine (which are present in living organisms) did not increased its antioxidant properties.

5.) To our knowledge, nothing was published about that: H2S and cysteine (which are present in living organisms) interacted with the extract and thus cleavage plasmid DNA in bell-shape concentration dependencies.

6.) To our knowledge, nothing was published about that: Extract, in the concentration dependent manner, inhibited DNA damage caused by polysulfides.

7.) To our knowledge, nothing was published about that: Looking for the components of the extract, which are responsible for the above described effects, we found that the extract components: Caffeic acid, chlorogenic acid and to less extent protocatechuic acid interacted with H2S and thus significantly increased their antioxidant properties.

8.) To our knowledge, nothing was published about that: The extract components: Caffeic acid, chlorogenic acid and to less extent protocatechuic acid interacted with cysteine and thus significantly increased their antioxidant properties.

9.) To our knowledge, nothing was published about that: The extract components: Caffeic acid, chlorogenic acid and to less extent protocatechuic acid interacted with glutathione and thus significantly increased their antioxidant properties.

10.) To our knowledge, nothing was published about that: Possible interaction of the extract components eleutheroside B or eleutheroside E with H2S, cysteine or glutathione did not increased their antioxidant properties.  

11.) To our knowledge, nothing was published about that: Caffeic acid, chlorogenic acid, protocatechuic acid and eleutheroside B (at high concentrations) inhibited DNA damage caused by polysulfides.

Reviewer 3 Report

 In this paper, the authors studied the influence of sulfide, cysteine, and glutathione on the antioxidant and DNA damaging properties of Acanthopanax senticosus (ASRE) and some of its selected phytochemical components.

The manuscript is, in general, clearly written. The scientific base and technical execution of the study are of good quality. The study is solid and the data, in general, support the conclusions. The theory is logical and experimental design is easy to follow and in general makes sense.
However, minor issues should be improved and corrected.

Abstract

Line 17: please, put the abbreviation for •cPTIO

Introduction

Well written but the aim of the study should be clear

Line 56: “we have studied” should be replaced by “we aimed to study”.

Line 59-65 the findings of the study should not be in the introduction. These should be considered in the conclusion.

Results

Resolution of figures could be improved

Discussion

Line 301: The present work was aimed at….. This sentence should be moved to the end of the introduction.

Conclusion

Line 359: starting from” The finding of the mutual in vitro effects…. Till the end of the discussion. It is better to be in a separate paragraph as a conclusion.

Author Response

Responses to the Reviewer 3. Manuscript ID: molecules-1872169

Thank you for your valuable remarks.

General comment 1.

English language and style are fine/minor spell check required
Answer 1.

We improved the English language style and checked the spelling.

General comment 2.

Does the introduction provide sufficient background and include all relevant references? Can be improved.

Answer 2.

In the revised manuscript: In introduction we provide more sufficient background included more references relevant to the research.

General comment 3.

Are the conclusions supported by the results? Can be improved.

Answer 3.

In the revised manuscript: We improved the conclusions supported by the results, see “Discussion”.

Comment 3.1.

Abstract Line 17: please, put the abbreviation for •cPTIO

Answer 3.1.

We corrected it.

Comment 3.2.

Introduction: Well written but the aim of the study should be clear

Line 56: “we have studied” should be replaced by “we aimed to study”.

Line 59-65 the findings of the study should not be in the introduction. These should be considered in the conclusion.

Answer 3.2.

We corrected it according to the suggestion.

Comment 3.3.

Results Resolution of figures could be improved

Answer 3.3.

We exchanged the figures with ones having 600 dpi resolution.

Comment 3.4.

Discussion Line 301: The present work was aimed at….. This sentence should be moved to the end of the introduction.

Answer 3.4.

We corrected it according to the suggestion. See the marked revised text.

Comment 3.5.

Conclusion Line 359: starting from” The finding of the mutual in vitro effects…. Till the end of the discussion. It is better to be in a separate paragraph as a conclusion.

Answer 3.5.

We corrected it according to the suggestion. See the marked revised text.

Round 2

Reviewer 2 Report

Accept in the present form